# Immunostimulatory Effects of Guanine-Quadruplex Topologies as Scaffolds for CpG Oligodeoxynucleotides

**DOI:** 10.3390/biom15010095

**Published:** 2025-01-10

**Authors:** Soumitra Pathak, Nguyen Bui Thao Le, Taiji Oyama, Yusuke Odahara, Atsuya Momotake, Kazunori Ikebukuro, Chiho Kataoka-Hamai, Chiaki Yoshikawa, Kohsaku Kawakami, Yoshihisa Kaizuka, Tomohiko Yamazaki

**Affiliations:** 1Research Center for Macromolecules and Biomaterials, National Institute for Materials Science (NIMS), 1-2-1 Sengen, Tsukuba 305-0047, Japan; pathak.soumitra@nims.go.jp (S.P.); jtef1940@tmd.ac.jp (N.B.T.L.); kataoka.chiho@nims.go.jp (C.K.-H.); yoshikawa.chiaki@nims.go.jp (C.Y.); kawakami.kohsaku@nims.go.jp (K.K.); kaizuka.yoshihisa@nims.go.jp (Y.K.); 2Graduate School of Life Science, Hokkaido University, Kita 10, Nishi 8, Sapporo 060-0808, Japan; 3JASCO Corporation, Hachioji 192-8537, Japan; taiji.oyama@jasco.co.jp; 4Department of Biotechnology and Life Science, Graduate School of Engineering, Tokyo University of Agriculture and Technology, Koganei 184-8588, Japan; ikebu@cc.tuat.ac.jp; 5Department of Chemistry, University of Tsukuba, Tsukuba 305-8571, Japan; s2220202@u.tsukuba.ac.jp (Y.O.); amomotak@chem.tsukuba.ac.jp (A.M.)

**Keywords:** CpG oligodeoxynucleotides, adjuvants, guanine-quadruplex, immune response, toll like receptor 9, macrophage cells, topology, serum stability, cellular uptake

## Abstract

Synthetic cytosine-phosphate-guanine oligodeoxynucleotides (CpG ODNs) are promising candidates for vaccine adjuvants, because they activate immune responses through the Toll-like receptor 9 (TLR9) pathway. However, unmodified CpG ODNs are quickly degraded by serum nucleases, and their negative charge hinders cellular uptake, limiting their clinical application. Our group previously reported that guanine-quadruplex (G4)-forming CpG ODNs exhibit enhanced stability and cellular uptake. G4 structures can form in parallel, anti-parallel, or hybrid topologies, depending on strand orientation, but the effects of these topologies on CpG ODNs have not yet been explored. In this study, we designed three distinct G4 topologies as scaffolds for CpG ODNs. Among the three topology, the parallel G4 CpG ODN demonstrated the highest serum stability and cellular uptake, resulting in the strongest immune response from macrophage cells. Additionally, we investigated the binding affinities of the different G4 topologies to macrophage scavenger receptor-1 and TLR9, both of which are key to immune activation. These findings provide valuable insights into the development of CpG ODN-based vaccine adjuvants.

## 1. Introduction

Innate immunity is activated when the immune system encounters pathogenic particles or pathogen-associated molecular patterns (PAMPs), which are present in many infectious microorganisms [1]. DNA motifs containing unmethylated cytosine-guanine dinucleotides (CpG) flanked by two purines at the 5′ end and two pyrimidines at the 3′ end are recognized by Toll-like receptor 9 (TLR9), leading to the production of interleukins (IL-6, IL-12, and IL-18), immunoglobulin M, interferon-γ, and tumor necrosis factor-α, thereby initiating an innate immune response. These CpG-rich motifs are approximately 20 times more common in microbial DNA than in mammalian DNA due to differences in their utilization frequency [2]. The scarcity of CpG motifs in mammalian DNA, coupled with cytosine methylation that inactivates these motifs, reflects evolutionary divergence, making CpG dinucleotide-containing DNA motifs potent PAMPs with strong immunostimulatory potential [3]. In the absence of microbial pathogens, synthetic oligodeoxynucleotides (ODNs) containing CpG motifs can mimic this immunostimulatory effect, positioning them as promising candidates for vaccine adjuvants [4].

However, the clinical application of CpG ODNs is limited by enzymatic degradation through serum nucleases, which target the phosphodiester (PD) bonds in these molecules. Various strategies have been employed to stabilize CpG ODNs for therapeutic use. For instance, substituting the PD bond’s double-bonded oxygen with sulfur to form a phosphorothioate (PT) bond has been shown to increase stability against enzymatic degradation [5]. Although this modification enhances stability, it also raises biosafety concerns. PT-modified CpG ODNs have been associated with prolonged local immune stimulation, which raises concerns about potential autoimmune responses [6,7], and cases of significant lymphadenopathy have been reported following administration [8].

Guanine-quadruplexes (G4s) represent a non-canonical secondary structure of single-stranded nucleic acids rich in guanine sequences. G4s are categorized as either intramolecular or intermolecular structures, with intramolecular G4s showing particular promise in pharmaceutical applications due to their monomeric nature and controllable structural properties [9]. Guanine residues within the G-tracts of a sequence form Hoogsteen hydrogen bonds with each other, creating G-quartets [10]. Multiple G-quartets can stack, and centrally located metal ions such as K^+^ and Na^+^ interact with the O_6_ position of guanine [11], generating a strong negative electrostatic potential that stabilizes the G-quadruplex structure under physiological conditions [12,13]. Our group previously leveraged the G4 structure as a scaffold for non-modified PD backbone-based CpG ODNs, achieving enhanced serum stability [14,15,16].

G-quadruplexes can be classified into three topologies—parallel, anti-parallel, and hybrid—based on the glycosidic bond angles (anti or syn) of the guanine residues in the G-tracts [17]. In parallel topology, the four guanine strands align in the same direction; in anti-parallel, two strands are oriented in one direction and two in the opposite; in hybrid topology, one strand has a different direction than the other three [18]. Factors influencing G-quadruplex folding into a specific topology include the type of metal ion present between G-quartets. For instance, K^+^ ions in human telomeric sequences promote a hybrid topology, while Na^+^ favors anti-parallel [19] and Ca^2+^ encourages parallel topology [20]. The length of loops between G-tracts also influences G4 topology, with shorter loops typically favoring a parallel structure [21,22]. Studies indicate that parallel G4s tend to adopt stable conformations, making them a preferred topology under physiological conditions [23,24].

Our previous work involved designing G4 CpG ODNs in hybrid and anti-parallel topologies to enhance their nuclease resistance and cellular uptake, which, in turn, augmented their immunostimulatory effects [14,25]. We further demonstrated that G4 CpG ODNs initially in hybrid topology could transition to parallel topology upon binding with the ligand L2G2-2M2EG-6OTD. This ligand-induced parallel topology increased nuclease resistance and cellular uptake but significantly reduced immunostimulatory activity compared to the original hybrid G4 CpG ODN [16].

Although G-quadruplexes show great potential as scaffolds for CpG ODNs, the impact of topological variation on immunological outcomes remains poorly understood. This study aims to elucidate the effect of G-quadruplex topology on immunostimulatory activity. We designed and characterized three G4 topologies—parallel, anti-parallel, and hybrid—incorporated with CpG ODNs. We then evaluated their topology in both extracellular and intracellular environments, as well as their nuclease resistance, cellular uptake, TLR9 interaction, and immunostimulatory effects. Our results demonstrate that the parallel topology is superior as a CpG ODN scaffold, with its high nuclease resistance and cellular uptake contributing significantly to enhanced immunostimulatory activity.

## 2. Materials and Methods

### 2.1. ODNs

All ODNs with PD backbones, listed in Table 1, were purchased from Eurofins Genomics (Tokyo, Japan) and were of HPLC grade. To assess cellular uptake, Cy5-labeled ODNs were used, with labeling at the 5′ end.

### 2.2. G4 Structure Formation

G4 structures were formed from randomly coiled ODNs following a previously published protocol from our group [25]. Briefly, all ODNs were diluted in Dulbecco’s phosphate-buffered saline (DPBS; Nacalai Tesque, Kyoto, Japan) containing 2.68 mM KCl, 137 mM NaCl, 1.47 mM KH_2_PO_4_, and 8.10 mM Na_2_HPO_4_. The ODN solutions were heated in a thermal cycler (PCR Thermal Cycler Dice Standard TP650, Takara Bio, Shiga, Japan) at 95 °C for 5 min to fully dissociate the structure and then cooled to 30 °C at a rate of 1 °C/min to induce G4 formation. The folded ODN solutions were stored at 4 °C until further use.

### 2.3. Circular Dichroism Spectroscopy

To investigate the topology of G4 CpG ODNs, Circular dichroism (CD) spectra were obtained in DPBS using a J-725 or J-1500 spectropolarimeter (JASCO, Tokyo, Japan) as previously described [15]. ODNs were prepared at a concentration of 2 µM in a 10 mm path-length quartz cell. G4 CpG ODN topologies were also analyzed under simulated early endosomal and lysosomal conditions using vesicle-mimicking buffers: an early endosomal buffer; a 60 mM potassium phosphate buffer containing 20 mM NaCl and 20% (*w*/*v*) PEG200 at pH 6.5, and a lysosomal buffer; and a 175 mM acetic acid buffer containing 60 mM KCl, 20 mM NaCl, and 20% (*w*/*v*) PEG200 at pH 4.5. G4 CpG ODNs were incubated in these buffers at 37 °C for 3 h before measuring their CD spectra. Melting curve analysis was performed using the baseline method [26] in DPBS buffer, and melting temperatures (T_m_) were calculated based on a previously reported method [25]. Principal component analysis (PCA) of the G4 CpG ODNs was conducted as outlined in our previous work [16]. A library of 30 reference CD spectra of G4 structures was used to create the PCA score plot, and then 95% confidence ellipses were created using the PCA scores 1 and 2 of each group of reference samples.

### 2.4. Polyacrylamide Gel Electrophoresis (PAGE)

PAGE was conducted with 1 mm thick, 4–20% pre-cast polyacrylamide gels (TEFCO, Tokyo, Japan) in Tris-borate-EDTA (TBE; 0.089 M Tris-Borate, 0.002 M EDTA, pH 8.3; Takara Bio, Shiga, Japan) buffer supplemented with 4 mM KCl at a constant voltage of 180 V for approximately 80 min, as previously described [15]. The electrophoresis was performed at 4 °C to prevent G4 CpG ODN conformational changes due to heat.

### 2.5. Size Exclusion Chromatography (SEC-HPLC)

SEC-HPLC analysis was performed using a Yarra SEC-2000 column (300 × 7.8 mm, 3 µm, Phenomenex, Torrance, CA, USA) with DPBS as the mobile phase. A 5 µL volume of 5 µM ODNs was injected, and the flow rate was set at 0.6 mL/min. Absorbance was detected at 260 nm to determine the elution time.

### 2.6. Purity Analysis via NMR

A ^1^H NMR spectrum was recorded using an AVANCE 600 spectrometer (Bruker Instruments, Inc., Bellerica, MA, USA). Folded G4 CpG ODNs at a concentration of 300 µM were dissolved in MilliQ water containing 10% D_2_O, 50 mM KCl, and 10 mM potassium phosphate buffer (pH 6.59). Spectra were acquired at a frequency of 600 MHz with a spectral width of 25 ppm, 8192 data points, a relaxation delay of 2 s, and 64 transients at 25 °C. A total of 1024 scans were performed, with water suppression achieved using the watergate method [27].

### 2.7. Serum Stability Assay

The serum stability of G4 CpG ODNs was evaluated following a previously described protocol [28]. Briefly, ODNs were incubated in 50% (*v*/*v*) fetal bovine serum (FBS; Sigma-Aldrich, St. Louis, MO, USA) at 37 °C for 0, 1, 2, 4, and 24 h. The amount of undegraded ODNs was quantified using PAGE.

### 2.8. Cell Culture

Mouse macrophage-like RAW 264 cells (RIKEN BioResource Center, Ibaraki, Japan) were cultured as described in our previous studies [15]. RAW264.7 cells with knocked-out macrophage scavenger receptor-1 (Msr-1) genes (RAW MSR-1 KO) were generously provided by Prof. Makiya Nishikawa of the Tokyo University of Science [29]. Cells were maintained in RPMI 1640 medium (Thermo Fisher Scientific, Waltham, MA, USA) supplemented with 10% (*v*/*v*) heat-inactivated FBS, 100 U/mL penicillin, and 100 µg/mL streptomycin. Cultures were incubated at 37 °C in a humidified atmosphere with 5% CO_2_.

### 2.9. Quantification of Immunostimulatory Activity

To achieve a cell density of 1 × 10^5^ cells/well, 190 µL of cell suspension (5.3 × 10^5^ cells/mL) was added to each well of a 96-well plate. After 18 h of incubation, 10 µL of ODN solution was added to the culture medium. Following 24 h of stimulation at 37 °C in a humidified 5% CO_2_ incubator, the supernatant was collected. Interleukin-6 (IL-6) levels were measured using a mouse IL-6 ELISA kit (Ready-Set-Go kit, Thermo Fisher, Waltham, MA, USA) following the manufacturer’s instructions.

### 2.10. Quantification of Cellular Uptake

Cells (4 × 10^5^ cells/well) were seeded in a 48-well plate and incubated for 18 h. The medium was then replaced with 200 µL of serum-free opti-MEM (Thermo Fisher Scientific) containing 0.5 µM Cy5-labeled G4 CpG ODNs. After 2 h, cells were harvested using a 0.5% (*w*/*v*) trypsin-0.2 mmol/L EDTA treatment, washed twice with phosphate-buffered saline (PBS), and collected by centrifugation at 500× *g* for 10 min. Cells were then resuspended in 500 µL PBS, and the mean fluorescence intensity (MFI) from Cy5 was quantified using a spectral cell analyzer (SP6800, Sony, Tokyo, Japan).

### 2.11. Confocal Microscopy

The localization of G4 CpG ODNs was examined via confocal microscopy (TCS SP5, Leica Microsystems, Wetzlar, Germany). Cells (2 × 10^5^) were cultured on a 35 mm ibiTreat dish with a 4-well silicon micro-insert (ibidi, Gräfelfing, Germany) for 24 h to form a cell layer. The medium was replaced with 10 µL of fresh medium containing 4 µM Cy5-labeled G4 CpG ODN. After 2 h, cells were washed twice with PBS. The MemBrite Fix Cell Surface Staining Kit (Biotium, Fremont, CA, USA) was used for cell membrane staining. Cells were washed twice with ice-cold PBS and fixed with 4% paraformaldehyde at room temperature for 10 min. After further washing with PBS, 10 µL of SlowFade Diamond Antifade Mountant with DAPI (Thermo Fisher Scientific) was added to each well to stain the nuclei. Cy5 signal quantification was performed on three randomly selected points in the confocal images.

### 2.12. Immunoprecipitation Assay

To quantify the binding of G4 CpG ODNs to TLR9, an immunoprecipitation assay was performed using the Dynabeads Protein G Immunoprecipitation Kit (Thermo Fisher Scientific). A total of 50 μL of Dynabeads solution was mixed with 200 μL of recombinant mouse TLR9Fc (mTLR9Fc, 2 μg/mL in PBS, R&D Systems, Minneapolis, MN, USA) and incubated at 25 °C for 20 min. The mTLR9Fc-immobilized Dynabeads were then washed twice with PBS to remove excess mTLR9Fc. G4 CpG ODNs were diluted to 10 μM in early endosomal buffer (pH 6.5) and added to the mTLR9Fc-immobilized Dynabeads, followed by a 30 min incubation, resulting in a final G4 CpG ODN concentration of 2 μM. The Dynabeads were washed with early endosomal buffer, and the G4 CpG ODNs bound to TLR9 were eluted using 10 μL of 50 mM glycine (pH 2.8). The G4 CpG ODN bound to TLR9 was analyzed by PAGE using 15% polyacrylamide gel in TBE buffer, and SDS-PAGE was performed in TG-SDS buffer using a 15% polyacrylamide gel to confirm the presence of mTLR9Fc.

### 2.13. Statistical Analysis

One-way analysis of variance (ANOVA) was used to assess statistical differences, followed by Tukey’s multiple comparisons test for comparisons between groups or Dunnett’s multiple comparisons test when comparing to a control group. Statistical analyses were performed using GraphPad Prism version 8.2.0 for Windows (GraphPad Software, Boston, MA, USA).

## 3. Results

### 3.1. Design and Characterization of G4 CpG ODNs

We designed three distinct G4 CpG ODNs with different topologies, each containing two ‘GTCGTT’ CpG motifs within the second loop. Previously, our group synthesized GD2_H, a G-quadruplex with a hybrid topology that includes two CpG motifs in the second loop [25]. In this study, we developed G4 CpG ODNs with parallel and anti-parallel topologies, using GD2_H as a template sequence. By modifying the nucleotide length in the loops of GD2_H, we created the mutants GD2_AP and GD2_P (Table 1). For GD2_AP, an adenine base was added to each side of the central loop containing the CpG motif. According to previous reports [30], a shorter loop region promotes a parallel G-quadruplex topology, so we designed GD2_P by deleting one thymine from the first and third loops of GD2_AP. To confirm molecularity, native PAGE was performed in the presence of 4 mM potassium ions in the running buffer. GD2_H (30-mer), GD2_AP (32-mer), and GD2_P (30-mer) displayed higher migration in polyacrylamide gel compared to single-stranded ODNs of similar length and linear structure (Figure 1A), suggesting compact three-dimensional structures. GD2_P showed the highest mobility among the three samples, while GD2_H and GD2_AP displayed distinct single bands. For GD2_P, an additional faint band around the 37 bp region was observed alongside the primary band. Further analysis using SEC-HPLC revealed a single peak for each sample, with elution times longer than those of linear samples, confirming compact structures (Figure 1B–D). The primary peak area for GD2_P was about 90%, indicating that the majority of GD2_P was present as a monomeric parallel topology.

Additionally, the formation of G-quadruplex DNA, GD2_P, was confirmed through characterization using ^1^H NMR. In the downfield-shifted portion of the NMR spectrum of GD2_P (Appendix A), imino proton signals were observed at the chemical shifts of ≳10 ppm, a range characteristic of guanine imino protons involved in hydrogen bond formation with their carbonyl oxygen atoms (i.e., NH−OC hydrogen bonds) [31]. The observation of the sharp imino proton signals is consistent with the formation of a G-quadruplex structure stabilized by three stacked G-quartets, indicating that the G-quadruplex adopts a relatively homogeneous conformation.

To confirm the G-quadruplex topology, we measured the CD spectrum in DPBS, which simulates physiological ion concentrations of the cell culture medium. At 37 °C in DPBS, GD2_H exhibited a CD spectrum with a negative peak around 240 nm, a broad positive peak around 260 nm, and another positive maximum around 290 nm (Figure 2A), characteristic of a hybrid G4 topology [32]. GD2_AP displayed a negative minimum around 260 nm and a positive maximum around 290 nm, indicating an anti-parallel topology [33]. Conversely, GD2_P showed a parallel topology with a negative minimum at 240 nm and a positive maximum at 260 nm [33]. All of the G4 CpG ODNs were found to maintain their respective topologies also at a room temperature of 25 °C (Appendix A). CD spectra were also obtained under conditions mimicking the intracellular environment. Although the three G4 CpG ODNs displayed different topologies in DPBS, all three exhibited spectra with a positive maximum of around 260 nm and a negative minimum of around 240 nm in early endosomal and lysosomal conditions (Figure 2B,C). A visual and subjective judgment is employed to determine the topology of G4 based solely on the shape of the CD spectrum. The topology of G4 nucleic acids can be evaluated objectively and quantitatively by using PCA based on CD spectra of G4 structures with known 30 structures [16,34]. GD2_H, GD2_AP, and GD_P were not within the ellipse because of the long random insertion in the second loop of G4, affecting the whole CD spectrum. PCA score plots confirmed that GD2_H, GD2_AP, and GD2_P formed hybrid, anti-parallel, and parallel structures, respectively, in DPBS (Figure 2D). Moreover, PCA indicated that all three G4 CpG ODNs adopted parallel topologies under conditions simulating early endosomes and lysosomes. These findings demonstrated that all three types of G4 CpG ODNs maintained their topology in the medium. After cell uptake, the topologies of GD2_H and GD2_AP changed into the parallel type within the endosome.

### 3.2. Effect of G4 Topology on the Immunostimulatory Activity of CpG ODNs

The immunostimulatory effects of G4 CpG ODNs were assessed by measuring IL-6 production in mouse macrophage-like RAW264 cells using ELISA. Following 24 h of incubation with the ODNs, cells treated with GD2_P produced IL-6 at levels four times higher than those treated with GD2_H and twice as high as those treated with GD2_AP, as shown in Figure 3. To confirm that IL-6 induction was due to the CpG motif in the second loop and not the G4 structure itself, we replaced the CG dinucleotide sequence with GC in all three G4 CpG ODNs, which formed GD2_H-GpC, GD2_AP-GpC, and GD2_P-GpC (Table 1). As a result, no IL-6 induction was observed in RAW264 cells treated with the modified G4 CpG ODNs with inverted CG sequences (Appendix A). CD spectra were analyzed to confirm whether the G4 GpC ODNs maintain their respective topologies even after the inversion of CG dinucleotide. GD2_H, GD2_AP, and GD2_P were shown to retain their hybrid, anti-parallel, and parallel topologies, respectively (Appendix A).

### 3.3. Effect of G4 Topology on Thermal Stability and Nuclease Resistance

The structural stability of G4 CpG ODNs with different topologies was evaluated through thermodynamic analysis using CD melting analysis. The temperature was increased at a rate of 1 °C/min over a range of 10 °C to 90 °C (Appendix A). The T_m_ of GD2_H, GD2_AP, and GD2_P were determined to be 46.6 °C, 45.2 °C, and 48.6 °C, respectively. These results indicate that the parallel structure GD2_P exhibits greater stability than the other topologies.

We then investigated the nuclease resistance of the three G4 CpG ODN topologies in FBS, which contains nucleases known to degrade unmodified CpG ODNs. As illustrated in Figure 4, linear ODNs degraded rapidly in DPBS containing 50% FBS, with no residual band visible after just an hour. In contrast, residual bands for the three G4 topologies were observed after 1, 2, and 4 h of incubation with FBS. Notably, after 24 h, the band for the parallel G4 (GD2_P) remained visible, whereas the bands for the hybrid and anti-parallel G4 structures were no longer detectable. Quantification of the ODN bands confirmed that the parallel G4 CpG ODN (GD2_P) displayed the highest stability among the three topologies (Figure 4D).

### 3.4. Effect of G4 Topology on Cellular Uptake of CpG ODNs

CpG ODNs activate the immune system by binding to TLR9 in endosomes [35]. For G4 CpG ODNs to induce an immune response in cells, they must first be internalized. We examined the cellular uptake of Cy5-labeled G4 CpG ODNs using flow cytometry. Histograms of fluorescence levels in the cells treated with the different G4 CpG ODN topologies are presented in Appendix A. Figure 5 displays the flow cytometry analysis of the RAW264 cells after 2 h of incubation with Cy5-labeled G4 CpG ODNs. Among the three G-quadruplex topologies, the parallel-type GD2_P exhibited comparatively higher cellular uptake, followed by GD2_AP and GD2_H. We also compared the uptake rate over time for each topology (Appendix A). Cellular uptake was detectable after 10 min, with the parallel GD2_P showing the highest uptake rate among all topologies.

Some ODNs bind non-specifically to cell membranes, remaining localized on the membrane [36]. To confirm internalization, we observed the localization of Cy5-labeled G4 CpG ODNs using confocal microscopy. As shown in Figure 6, all Cy5-labeled G4 CpG ODNs (visualized in red) displayed internalization within RAW264 cells. The Cy5-labeled G4 CpG ODNs localized in the cytoplasm in a punctate pattern, confirming their presence in intracellular vesicles. Quantification of the Cy5 signal revealed that the parallel GD2_P had the highest internalization level among the three topologies (Appendix A).

### 3.5. Investigation of the Uptake Receptor for G4 CpG ODNs

We also explored the cellular uptake pathways for the different G4 CpG ODN topologies. Cell membranes contain various receptors and proteins that facilitate the uptake of DNA nanostructures [37,38]. Among these, the macrophage scavenger receptor-1 (MSR-1) receptor, encoded by the MSR-1 gene, has been identified as a key facilitator for the uptake of phosphodiester DNA in macrophage cells [29,39,40]. We investigated whether small DNA structures like G-quadruplexes are also internalized by immune cells through MSR-1, particularly when CpG motifs are incorporated.

As shown in Figure 7A, all three G4 CpG ODN topologies demonstrated reduced uptake in RAW MSR-1 knockout (KO) cells compared with RAW264 cells. The parallel-type GD2_P displayed a twofold reduction in internalization in RAW MSR-1 KO cells. We further assessed whether this reduced uptake affected cytokine secretion. MSR-1 KO cells showed no cytokine secretion in response to any of the three G4 CpG ODN topologies, in contrast to RAW264 cells (Figure 7B). These results indicate that the MSR-1 receptor is involved in the uptake of G4 CpG ODNs and plays a crucial role in immune activation in RAW cells.

### 3.6. TLR9 Affinity of G4 CpG ODNs and Immune Response

Following cellular uptake, CpG ODNs are directed to endosomes [41]. To examine the binding affinity between TLR9 and the CpG motifs located in the loop region (Figure 2E–G) of G4 CpG ODNs, an immunoprecipitation assay was conducted under endosome-mimicking conditions at pH 6.5 (Figure 8). Results showed that all three G4 CpG ODN topologies bound to TLR9. Quantitative analysis of fluorescence intensity in the ODN bands revealed similar levels of TLR9 binding affinity among the tested G4 CpG ODNs.

## 4. Discussion

The immune response mediated by CpG ODNs through recognition by the TLR9 receptor in immune cells presents a promising strategy for vaccine adjuvant development [42]. However, natural linear CpG ODNs suffer from low nuclease resistance, limiting their effectiveness. Although phosphorothioate-modified CpG ODNs offer increased stability, they raise safety concerns. To address this, our group previously developed G4 CpG ODNs as an alternative, avoiding safety risks while still activating immune responses in immune cells [14,15,25,43]. G4 CpG ODNs have shown higher immunostimulatory effects compared to linear CpG ODNs, primarily due to their enhanced stability in the presence of serum nucleases, making them a viable option for vaccine adjuvant applications.

G-quadruplexes can adopt three distinct topologies—parallel, anti-parallel, and hybrid—based on loop orientation [30]. However, the specific impact of these topologies on the immunostimulatory activity of G4 CpG ODNs remains unclear. This study aimed to design CpG ODNs with G-quadruplex scaffolds in these three topologies and to investigate how topology influences immunostimulatory effects. Our findings indicate that parallel G4 CpG ODNs exhibit superior nuclease resistance and cellular uptake compared to other topologies, which likely explains their enhanced immunostimulatory effects in macrophage cells.

Among all tested topologies, parallel G4 CpG ODNs triggered the strongest immunostimulatory response, as evidenced by the highest IL-6 secretion in RAW264 cells (Figure 3). To understand this enhanced response, we examined the stability of the ODNs in cellular media with 50% serum, reflecting physiological serum concentrations. Previous studies, such as that by Luu et al. [19], have shown that human telomeric sequences in Na^+^ solutions form anti-parallel G4 structures, while in K^+^ solutions, they adopt a more compact parallel topology. This compact structure likely contributes to the parallel G4 CpG ODN’s superior resistance to nuclease degradation, remaining stable even after 24 h of incubation (Figure 4D). This observation aligns with our previous work, which demonstrated that ligand-induced parallel G4 CpG ODNs retain high stability in serum [16]. After cellular uptake, inside the endosome, all of the G4 CpG ODNs of different topologies fold themselves into parallel topology due to the collective effect of molecular crowding and higher K^+^ concentration [44,45].

Cy5-labeled parallel G4 CpG ODNs exhibited the highest cellular uptake in RAW cells (Figure 5). The compact structure of the parallel G4 CpG ODN likely facilitates greater cellular uptake due to its smaller molecular profile [46]. Previous research has shown that G4 structures generally display enhanced uptake in cancer cells than non-G4 sequences, with membrane proteins involved in this uptake [47,48]. The higher serum stability of parallel G4 CpG ODNs likely contributes to their availability in cellular media for internalization. Additionally, the strong binding affinity to receptors such as MSR-1 may further enhance uptake into macrophage cells.

In the present study, experiments with MSR-1 knockout RAW cells showed reduced uptake of G4 CpG ODNs and diminished IL-6 secretion, with the most substantial reduction observed in the parallel topology (Figure 7). While various membrane proteins, including MAC-1, AGER, MSR-1, MNAB, DEC205, and MRC1, are involved in DNA uptake, only MAC-1 and MSR-1 are found in immune cells [39,49,50,51]. Given the role of MSR-1 as a macrophage and dendritic cell surface receptor [52,53], all G4 CpG ODN topologies possibly utilize MSR-1 for uptake and are subsequently directed to endosomes via clathrin-mediated endocytosis (CME) [54]. The reduced uptake and cytokine induction in MSR-1 knockout cells (Figure 7A,B) suggest that additional receptors may aid G4 CpG ODN uptake. Previous studies reported that the mannose receptor can target CpG oligonucleotides and internalize them in macrophage cells [51]. TRPC3/C6/C7 has also been reportedly involved in the uptake of antisense oligonucleotides [55]. Thus, other receptors apart from MSR-1 are possibly involved in RAW264 cells for internalizing G4 CpG ODNs. However, without MSR-1, internalized G4 CpG ODNs may not efficiently sort into endosomes, preventing TLR9 engagement and resulting in negligible IL-6 secretion (Figure 7B).

In summary, this study provides insights into the immunostimulatory mechanisms of G4 CpG ODNs. These molecules bind to MSR-1 receptors on macrophage cell surfaces, with parallel G4 CpG ODNs showing the highest affinity for receptor binding and subsequent internalization. Upon cellular entry, G4 CpG ODNs are trafficked to endosomes via the CME pathway, where all topologies converge into the parallel form, enabling TLR9 binding without topological preference. TLR9 engagement initiates downstream signaling, leading to IL-6 secretion [56]. Given its increased serum stability and cellular internalization, the parallel G4 CpG ODN stimulates a more robust IL-6 response than other G4 topologies.

## 5. Conclusions

We synthesized G4 CpG ODNs with three distinct topologies and examined their pathway from entry into cellular media to immune response activation, focusing on their interactions with different G4 CpG ODN topologies. Parallel G4 CpG ODNs demonstrated the highest serum stability, strongest binding affinity to MSR-1, and greatest cellular uptake, resulting in the most robust immune response in macrophage cells. These findings offer valuable insights for designing G-quadruplex-based CpG ODNs as vaccine adjuvants with improved stability and enhanced usability.

## Figures and Tables

**Figure 1 biomolecules-15-00095-f001:**
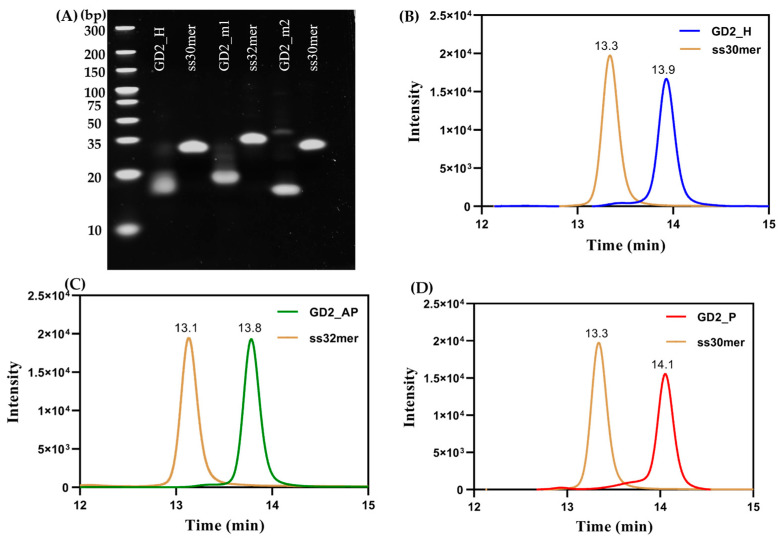
Molecularity analysis of G4 CpG ODNs. (**A**) Polyacrylamide gel electrophoresis analysis of G4 CpG ODNs was performed in a 4–20% gradient polyacrylamide gel in tris-borate EDTA buffer with 4mM KCl as supplement. (**B**–**D**) Size exclusion HPLC chromatograms of G4 CpG ODNs.

**Figure 2 biomolecules-15-00095-f002:**
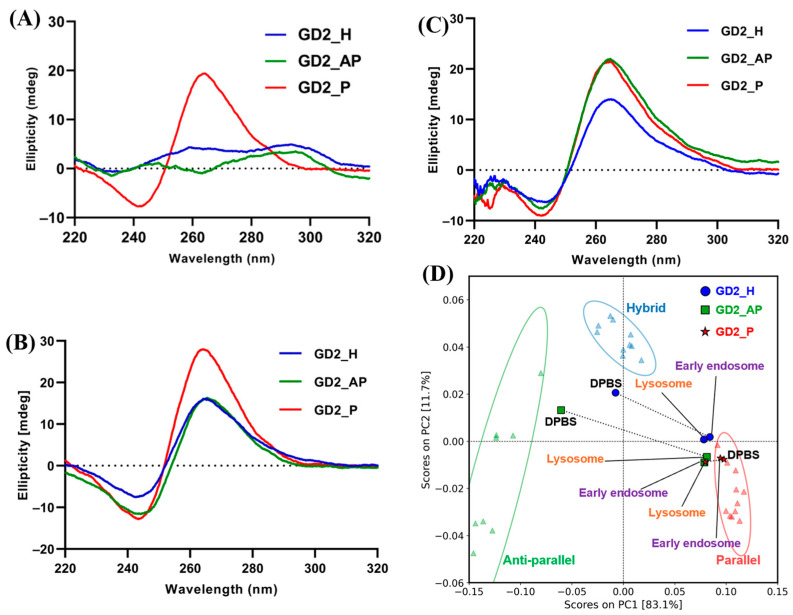
Circular dichroism (CD) spectrum of G4 CpG ODNs (**A**) in DPBS at 37 °C, (**B**) in early endosome mimicking buffer, and (**C**) in lysosome mimicking buffer at 37 °C. (**D**) Principal component analysis (PCA) score plots of G4 CpG ODNs. A library of 30 reference CD spectra of G4 structures was used to create the PCA score plot (shown as triangles). PCA score plot of GD2_H, GD2_AP, and GD_P in DPBS, early endosome, and lysosome mimicking buffer marked with blue circles, green squares, and red stars, respectively. The ellipses indicate 95% confidence limits of respective topologies. Schematic diagram of the G4 CpG ODNs, (**E**) GD2_H, (**F**) GD2_AP, and (**G**) GD2_P. Base pairs present on the red line are in the loop region, and G-tracts are depicted in green.

**Figure 3 biomolecules-15-00095-f003:**
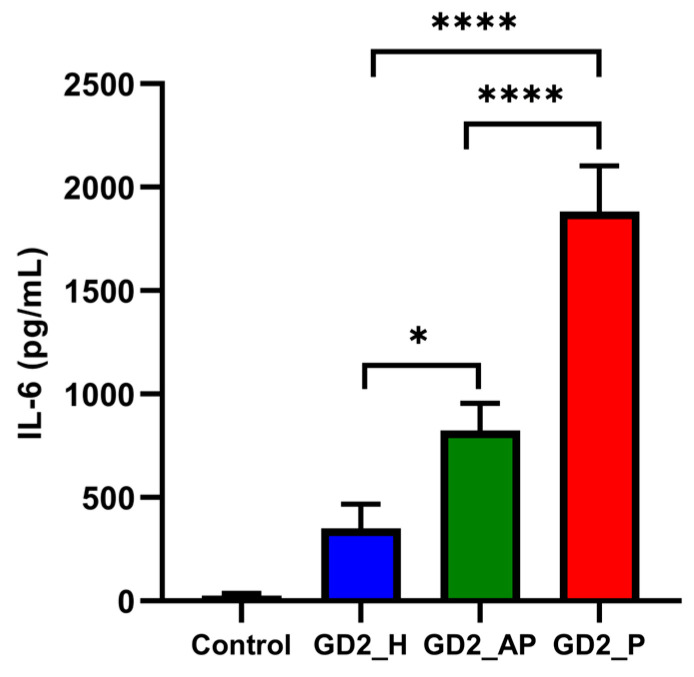
Cytokine induced by 4 μM of G4 CpG ODNs in mouse macrophage RAW264 cells after 24 h incubation. DPBS was used as control. Data are presented as means ± SD (*n* = 5). In the graph, **** *p* < 0.0001 and * *p* < 0.05 (one-way ANOVA, Tukey’s multiple comparisons test for comparison with other groups).

**Figure 4 biomolecules-15-00095-f004:**
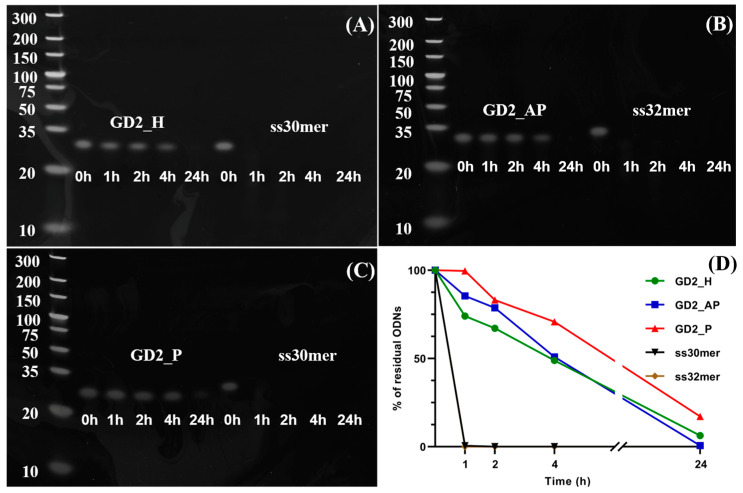
Serum stability of three types of G4 CpG ODNs. After incubating G4 CpG ODNs with 50% FBS, samples were run in PAGE to visualize residual ODNs. (**A**) Hybrid GD2 with ss30mer of same length, (**B**) anti-parallel GD2_AP and ss32mer of same length, and (**C**) parallel GD2_P and ss30mer of same length. Bands of DNA ladders are in bp. (**D**) Correlation between serum treatment time and residual ODN level.

**Figure 5 biomolecules-15-00095-f005:**
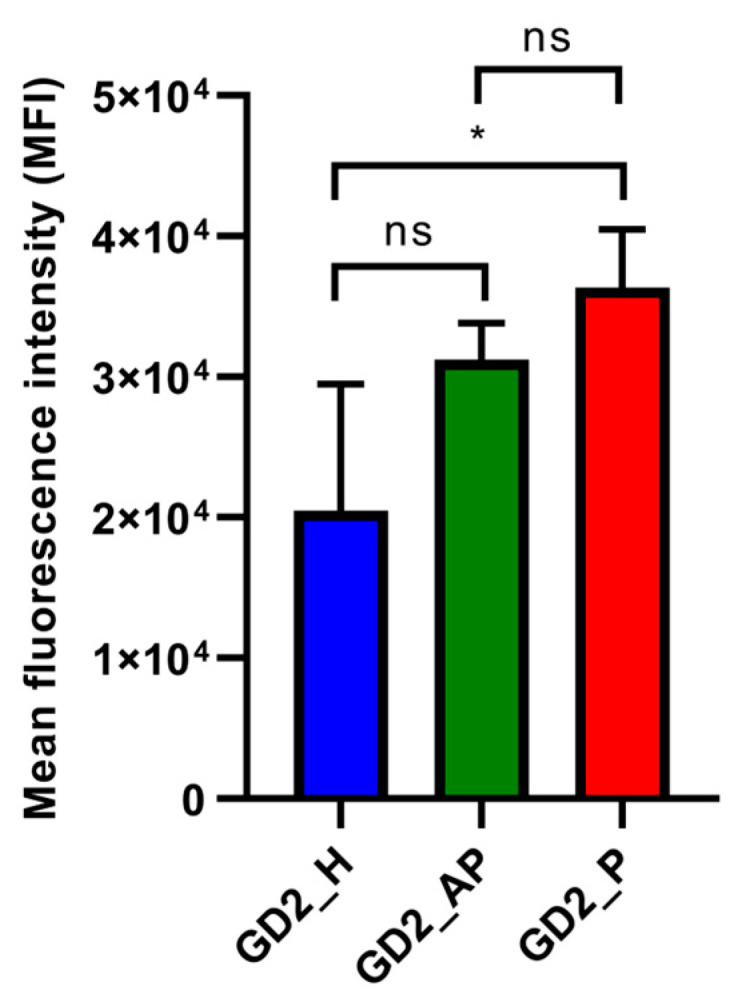
Cellular uptake of G4 CpG ODNs in RAW264 cells. Cells were incubated with Cy5-labeled G4 CpG ODNs for 2 h before quantification. Data are presented as means ± SD (*n* = 3). In the graph, * *p* < 0.05, and ns (not significantly different) means *p* > 0.05 (one-way ANOVA, Tukey’s multiple comparisons test for comparison with other groups).

**Figure 6 biomolecules-15-00095-f006:**
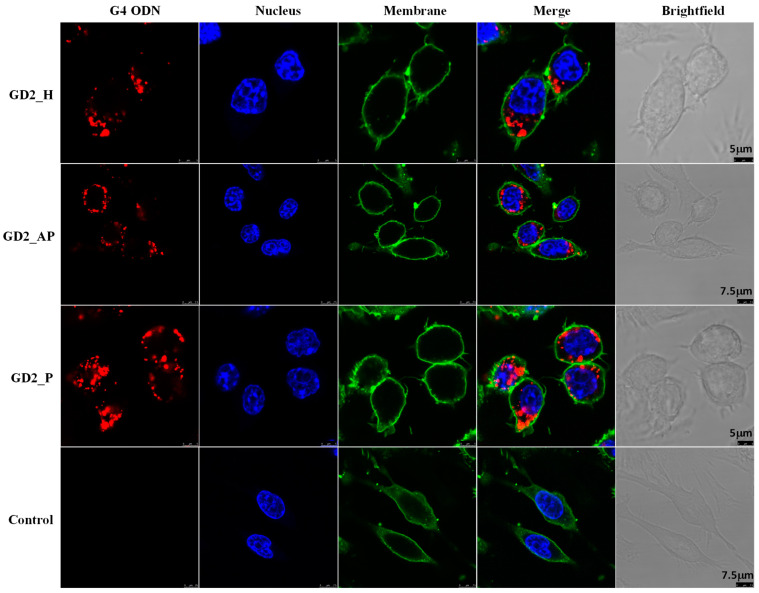
Localization of Cy5 labeled G4 CpG ODNs in RAW264 cells. The cells were incubated with 0.5 μM of ODNs for 2 h and then observed under confocal microscopy. No treatment RAW264 cells were used as a control. The CpG ODNs are marked with Cy5 (red). The cell membranes and nuclei are stained with MemBrite (green) and DAPI (blue), respectively.

**Figure 7 biomolecules-15-00095-f007:**
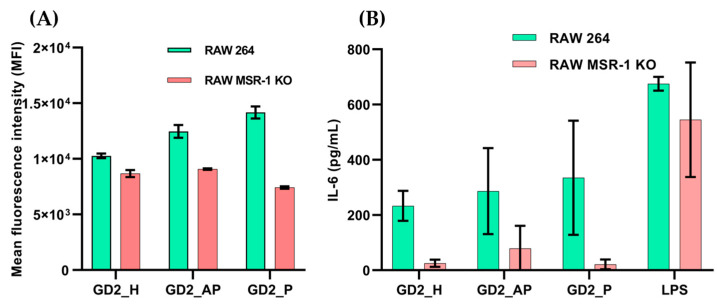
Studying receptor involved in cellular uptake of G4 CpG ODNs in RAW MSR-1 KO cells. (**A**) Cellular uptake of Cy5-labeled ODNs and (**B**) IL-6 secretion quantified by ELISA upon stimulation with G4 CpG ODNs. RAW264 cells were used as control. LPS was used as positive control for cell stimulation in ELISA.

**Figure 8 biomolecules-15-00095-f008:**
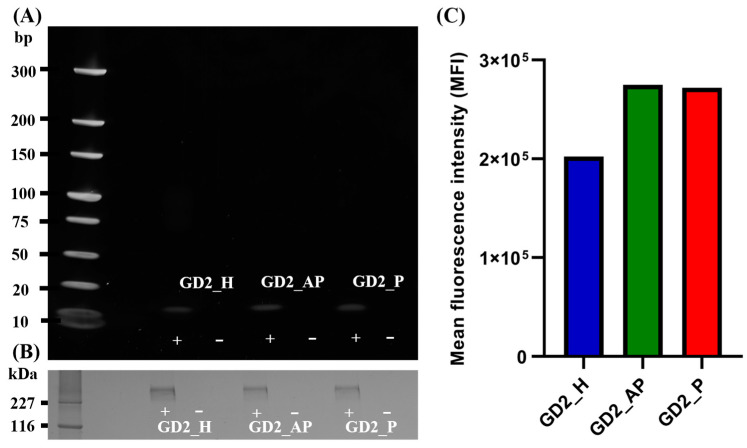
Binding between G4 CpG ODNs and mTLR9 using immunoprecipitation assay at pH 6.5. ‘−’ and ‘+’ sign signify absence of presence of mTLR9 Fc chimera in the sample, respectively. (**A**) Polyacrylamide gel (10–20%) visualized after staining with SBYR gold and running in TG buffer to observe DNA bands. (**B**) mTLR9 Fc chimera bound to protein G was visualized after running the sample in 15% polyacrylamide gel in TG-SDS buffer and stained in CBB stain. (**C**) Quantification of ODN bands visualized in PAGE.

**Table 1 biomolecules-15-00095-t001:** Sequence of oligonucleotides used in this study.

Name	Sequence (5′-3′)	Length (bp)
GD2_H	**GGG**TT**GGG**GTCGTTTTGTCGTT**GGG**TT**GGG**	30
GD2_AP	**GGG**TT**GGG**AGTCGTTTTGTCGTTA**GGG**TT**GGG**	32
GD2_P	**GGG**T**GGG**AGTCGTTTTGTCGTTA**GGG**T**GGG**	30
GD2_H-GpC	**GGG**TT**GGG**GTGCTTTTGTGCTT**GGG**TT**GGG**	30
GD2_AP-GpC	**GGG**TT**GGG**AGTGCTTTTGTGCTTA**GGG**TT**GGG**	32
GD2_P-GpC	**GGG**T**GGG**AGTGCTTTTGTGCTTA**GGG**T**GGG**	30
ss30mer	GTCGTTTTGTCGTTTTGTCGTTTTGTCGTT	30
ss32mer	GTCGTTTTGTCGTTTTGTCGTTTTGTCGTTTT	32

G-tracts are marked in bold, CpG/GpC motifs are underlined, and changes in GpC motifs are colored in red.

## Data Availability

The original contributions presented in this study are included in the article/Appendix A. Further inquiries can be directed to the corresponding author.

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
