# Peer review of "Immunostimulatory Effects of Guanine-Quadruplex Topologies as Scaffolds for CpG Oligodeoxynucleotides"

_biomolecules, 2025, doi:10.3390/biom15010095_

Round 1
Reviewer 1 Report
Comments and Suggestions for Authors
The authors present a clear and well designed study that I found very interesting and that can be of interest of a wide audience. The aim is to connect the topology of especially designed G4 oligos to their cellular activity. The authors test in a detailed way the immunostimulatory mechanisms of G4 CpG ODNs and highlight Structure-Activity Relationships (SAR). In particular, the role of a parallel topology is found to be major. On the whole, it is a nice work, the different parts are well equilibrated, the literature is sound, the comments/discussion of the results is robust.
I therefore suggest to publish this paper. I would ask the authors to just take into account the minor comments listed below.
Line 102/Table 1 "CpG/GpC motifs are highlighted in grey." This is very hard to be seen, somewhat better for the G-tracts. Colours (red, blue...) or underlines should be used.
Line 261/Figure 2 All this figure is not so clear. First, I would always use the same GD2_AP and GD2_P acronym all along the text, because changing it is misleading. i understand the authors want first to call oligo in a general way, and then change the acronym once the topolygy is confirmed..is it so? Still it seems me not needed. In that way, Figure 2 will have homogeneous legends.
"The ellipses indicate 95% confidence limits of respective topologies". Please explain better how were these ellipses calculated. The colours in Panel D are mislesding.The ellipse of parallel should have same colour of the parallel oligo, and the same hold for the other two topologies. Why is GD2_AP in DPBS not inside the anti-parallel ellipse? On the whole this panel (D) is more confusing than adding something to the very clear CD results...
Lines 267-268 "To confirm that IL-6 induction was due to the CpG motif in the 267 second loop and not the G4 structure itself, we replaced the CG dinucleotide sequence with GC in all three G4 CpG ODNs." This sentence is not totally clear, what is exactely the sequence and how long is it? Are they those in Table 1? If yes, please cite explicitly. How can we be sure that they do not form any type of secondary structure? To such a result, it would be great to have the CD spectra of these sequences.
Line 300 "GD2_P exhibited the highest cellular uptake among the three G-300 quadruplex topologies, followed by GD2_AP and GD2_H." This should be written in a less assaertive way, given that Figure 5 reports "ns" two times and just * is present...
Supplementary Figure S5 Why is the error of GD2_H at 120 min so high and different with respect to all other tests?
Supplementary Figure S2-S5-S6 Add ANOVA test also here? This is important in particular for S6 to demostrate that GD2_P had the highest internalization level among the three topologies (line 322).
Line 355 Figure 8C "Quantitative analysis of fluorescence intensity in the ODN bands revealed no significant difference in TLR9 binding among the 356
three topologies". This does not seem to be demostrated given that Figure 8C has no error bars/ANOVA test.
Figure S7 It seems strange that this so important picuture to undestrand the paper is put here in the end of it and as SI. Please transfer it in the main text, at its beginning.
Reviewer 2 Report
Comments and Suggestions for Authors
Please find attached file

Reviewer 3 Report
Comments and Suggestions for Authors
The manuscript submitted by S. Pathak and co-authors is interesting and well written. However, I have few comments or questions listed below.
1. Did Authors performed any studies on the numbers of molecules that are forming G-quadruplexes in endosome and lysosome mimicking conditions? The change in the CD spectra can be observed, which points to the change of the topology. There is no evidence provided on molecularity of those G4 in comparison to DPBS conditions. Additional experiments could provide more data regarding this topic.
2. What is the point of PCA presented in Figure 2? The same conclusions that were obatined based on the CD spectras alone were repeated based on PCA. Is there any other reason PCA should be there?
3. I would suggest addition of scheme presenting proposed structures formed by studied G4 CpG ODNs.
Round 2
Reviewer 2 Report
Comments and Suggestions for Authors
The Yamazaki et al. are exploring G4 structure as scaffold for CpG ODNs to trigger the immune response by stimulating endosomal Toll-like receptor (TLR) due to their better resistance to nuclease degradation, thereby enhancing their clinical applications in comparasion with unstructured ODNs. The presented version of manuscript after major revision has been improved significantlynot only in editorial term, but also contains corrected experiments nd answers to my doubts. Therefore, I recommend manuscript to be published.